# Caffeic Acid Phosphanium Derivatives: Potential Selective Antitumor, Antimicrobial and Antiprotozoal Agents

**DOI:** 10.3390/ijms25021200

**Published:** 2024-01-18

**Authors:** Miloš Lukáč, Lívia Slobodníková, Martin Mrva, Aneta Dušeková, Mária Garajová, Martin Kello, Dominika Šebová, Martin Pisárčik, Marián Kojnok, Andrej Vrták, Elena Kurin, Silvia Bittner Fialová

**Affiliations:** 1Department of Chemical Theory of Drugs, Faculty of Pharmacy, Comenius University Bratislava, Odbojárov 10, 832 32 Bratislava, Slovakia; lukac@fpharm.uniba.sk (M.L.); pisarcik@fpharm.uniba.sk (M.P.); mariankojnok219@gmail.com (M.K.); andrej1616@gmail.com (A.V.); 2Institute of Microbiology, Faculty of Medicine, Comenius University Bratislava, University Hospital in Bratislava, Sasinkova 4, 811 08 Bratislava, Slovakia; livia.slobodnikova@fmed.uniba.sk; 3Department of Zoology, Faculty of Natural Sciences, Comenius University Bratislava, Mlynská Dolina, Ilkovičova 6, 842 15 Bratislava, Slovakia; martin.mrva@uniba.sk (M.M.); aneta.dusekova@gmail.com (A.D.); maria.garajova@uniba.sk (M.G.); 4Department of Pharmacology, Faculty of Medicine, P.J. Šafárik University, Trieda SNP 1, 040 11 Košice, Slovakia; martin.kello@upjs.sk (M.K.); dominika.sebova@student.upjs.sk (D.Š.); 5Department of Pharmacognosy and Botany, Faculty of Pharmacy, Comenius University Bratislava, Odbojárov 10, 832 32 Bratislava, Slovakia; elena.kurin@uniba.sk

**Keywords:** caffeic acid, phosphanium salts, anticancer, antimicrobial, anti-*Acanthamoeba* activity

## Abstract

Caffeic acid (CA) is one of the most abundant natural compounds present in plants and has a broad spectrum of beneficial pharmacological activities. However, in some cases, synthetic derivation of original molecules can expand their scope. This study focuses on the synthesis of caffeic acid phosphanium derivatives with the ambition of increasing their biological activities. Four caffeic acid phosphanium salts (CAPs) were synthesized and tested for their cytotoxic, antibacterial, antifungal, and amoebicidal activity in vitro, with the aim of identifying the best area for their medicinal use. CAPs exhibited significantly stronger cytotoxic activity against tested cell lines (HeLa, HCT116, MDA-MB-231 MCF-7, A2058, PANC-1, Jurkat) in comparison to caffeic acid. Focusing on Jurkat cells (human leukemic T cell lymphoma), the IC_50_ value of CAPs ranged from 0.9 to 8.5 μM while IC_50_ of CA was >300 μM. Antimicrobial testing also confirmed significantly higher activity of CAPs against selected microbes in comparison to CA, especially for Gram-positive bacteria (MIC 13–57 μM) and the yeast *Candida albicans* (MIC 13–57 μM). The anti-*Acanthamoeba* activity was studied against two pathogenic *Acanthamoeba* strains. In the case of *A. lugdunensis*, all CAPs revealed a stronger inhibitory effect (EC_50_ 74–3125 μM) than CA (>10^5^ µM), while in *A. quina* strain, the higher inhibition was observed for three derivatives (EC_50_ 44–291 μM). The newly synthesized quaternary phosphanium salts of caffeic acid exhibited selective antitumor action and appeared to be promising antimicrobial agents for topical application, as well as potential molecules for further research.

## 1. Introduction

Caffeic acid (3-(3,4-dihydroxyphenyl)-2-propenoic acid) is formed by hydroxylation reactions from cinnamic acid in the shikimate biosynthetic pathway. In nature, it occurs almost exclusively in the more energy-efficient E isomer (trans configuration of double bound) [1]. The distribution of CA in the plant kingdom is extensive. It is one of the most abundant hydroxycinnamic acids in fruits, e.g., berries, and it might be found in vegetables (e.g., potatoes, carrots, artichokes, olives), in coffee, wine, tea, or propolis [2,3,4]. It can be found solo in simple monomeric form (esters, glycosides, amides) or as a part of a complex form of many other molecules–caffeic acid derivatives (dimers, trimers, flavonoid derivatives, bound to proteins and other polymers in the plant’s cell walls) [3]. CA has undergone extensive research for possible health-promoting uses in human and veterinary medicine and not only that but also in the field of the food and cosmetics industry. It is a substance of ubiquitous occurrence. In plants, especially with ferulic acid, it prevents seed germination [5]. CA is a fascinating molecule for preparing semi-synthetic derivatives. In many scientific works, CA and its derivatives have attracted considerable attention due to their various biological and pharmacological activities, including antioxidative [6,7,8], antimicrobial [9], and anticancer activity [4].

Recent research on CA has revealed many different molecular targets and pathways explaining the mechanism of action of CA and its derivatives [4,9,10]. Khan and colleagues reviewed the antimicrobial properties of different caffeic acid derivatives. Most of the tested molecules exhibited antimicrobial activity against Gram-positive and Gram-negative bacteria or fungi and were active against common viruses such as Herpes Simplex Virus type 1 or Influenza A virus. Glycosylated derivatives of CA reduced in silico viral load and infection period of virus SARS-CoV-2. Modifying caffeic acid has brought new molecules (derivatives/analogs) that displayed potential antimicrobial activity toward different types of pathogens [9]. Furthermore, the combination with antibiotics has also proved to be advantageous. The chitosan–caffeic acid conjugate in combination with conventional drugs, such as erythromycin, lincomycin, or tetracycline, revealed the synergistic antibacterial action. It is suggested that combinations of drugs with caffeic acid or caffeic acid derivatives may restore the full antibacterial activity of old commercial antibiotics in the fight against antibiotic-resistant bacteria [9,11,12].

Not only bacteria, fungi, and viruses are targets for researching new antimicrobial drugs. Acanthamoeba infections are severe health complications and life-threatening diseases. The currently available anti-acanthamoeba drugs possess many side effects, and the whole treatment is complicated by recurrent infections, inadequate treatment outcomes, and health complications [13,14]. A recently published study of the pharmacological properties of new derivatives of rosmarinic acid, also a caffeic acid derivative, shows potential effects against *A. quina* and *A. lugdunensis* [15]. Compounds of natural origin with low side effects should be welcome in the fight against pathogenic acanthamoebae.

Caffeic acid has possible anticancer activity in certain human cancers, possessing the ability to affect different molecular pathways of tumor cell proliferation [4], migration, and invasion with the potential to decrease metastases [16]. CA can improve the response of cancer cells to chemotherapy and sensitize them to chemotherapy-mediated cell death [4]. On the other hand, CA has very poor bioavailability results from its physicochemical and pharmacokinetic limitations, such as low water solubility, which are associated with unsatisfactory results after application [17]. For example, 300 mL of red wine provided about 2.8 mg of caffeic acid, but 1 h after its consumption, plasma levels of caffeic acid were as low as 4.9 ± 0.3 ng/mL (approx. 27.2 nM) [18]. Therefore, this pilot study aimed to modify caffeic acid by phosphanium groups to change its physicochemical properties to become more active in both eucaryotic and procaryotic cells. The beneficial surfactant-like properties rising from the phosphanium chain annexation could solve the hydrophilic CA’s low bioavailability and improve its biological effects. 

## 2. Results and Discussion

Prepared phosphanium salts derived from caffeic acids represent quaternary cationic amphiphilic compounds. The CAP structures are depicted in Figure 1. The compounds are structurally related to compounds we have published previously [15] and were derived from rosmarinic acid. Rosmarinic acid belongs to polyphenolic compounds and has a similar structure to caffeic acid. In this way, derivates were prepared with the aim of easier passage through biological membranes and changing the lipophilicity of the bioactive compound [15,19]. Based on previous experience with rosmarinic acid derivatives [15], the compounds were deliberately prepared with acetylated phenolic groups to increase the solubility of CAP in water.

The CAPs were prepared by three-step synthesis. Its reaction scheme is shown in Figure 1. Acetylation of caffeic acid was carried out by reaction with acetic anhydride in the presence of an organic base. The reaction yields were comparable to the literature data [20,21]. Phenolic groups were protected by means of an acetylating agent. This protection was essential for the successful completion of the second step of the synthesis, which was the preparation of esters. The esters of acetylated CA were prepared by a reaction of potassium carboxylate with α,ω-dibromoalkanes. The reaction is most often carried out in DMF, while various alkali metal carbonates can be used as the base [22]. Acetyl groups attached to the phenolic groups of CA prevented the side reaction with α,ω-dibromoalkane, and ether formation. For example, a similar reaction was used in the preparation of alkylphosphocholines derived from benzethonium chloride (Scheme 4 in [23]). The last step of the synthesis was the quaternization of compounds **II**–**V** with triphenylphosphane.

The physicochemical properties of the compounds were investigated by the measurements of surface tension of their aqueous solutions. The plots of surface tension vs. log concentration curves of CAPs are shown in Figure 2. The critical micelle concentration (cmc), the surface tension value at the cmc (γ_cmc_), and the surface area per head group at the surface saturation (A_cmc_) were determined. The results are summarized in Table 1.

The cmc values of CAPs decrease with the lengthening of the connecting chains between the ester group and the phosphanium cation of the compounds. It is related to the increase in the lipophilicity of the compounds depending on the increasing number of carbon atoms in the connecting chain. The values of γ_cmc_ are very similar for compounds **CAP 6**, **CAP 8**, and **CAP 10**. Only the compound **CAP 12** has a higher value of γ_cmc_. This is related to a less close arrangement of the molecules of the amphiphilic compound at the water-air interface. This is reflected in its value of A_cmc_. Compound **CAP 12** occupies the largest surface area at the water-air interface. This is related to the greatest flexibility of the connecting chain between the ester group and the phosphanium cation of the compound compared to other CAPs.

### 2.1. MTS Cell Proliferation/Viability Assay

As it is known, CA and its derivatives are involved in cancer cell death pathways by targeting apoptosis by increasing ROS levels and weakening mitochondrial function. Consequently, a reduction in growth and survival of the carcinoma cells can occur [4]. In our study, using the colorimetric MTS assay, the inhibitory effect of the studied substances was determined on several cancer cell lines from different human tissue origins (cervix, colon, breast, skin, pancreas, and blood). It is known that caffeic acid induces apoptosis and cell cycle arrest (S and G0/G1 phase) [24] and inhibits migration in breast cancer MDA-MB-231 cells [25]. Apoptosis, after caffeic acid treatment, also induces in other human breast cancer MCF-7 cells via an intrinsic signaling pathway [26]. The proliferation of colorectal cancer cells (HT-29 and HCT-116) is suppressed by caffeic acid [27]; thus, it seems that caffeic acid and its derivatives are promising candidates in cancer treatment research.

In the present study, we confirmed that CA and newly synthesized derivatives of CA inhibit the metabolic activity/growth of different human cancer cell lines, as can be seen in Table 2, Figure 3 and Figure 4. From the results, it is obvious that the compounds **CAP 10** and **CAP 12** exhibited the most significant inhibitory effects on almost the whole spectrum of tested cancer cell lines (except for PANC-1), with an IC_50_ value ranging from 0.9–9.5 μM (Table 2). Compound **CAP 8** also showed an inhibitory effect ranging from IC_50_ 4.8–9.4 μM; however, in cell lines MCF-7 and PANC-1, the inhibitory activity on cell metabolism was lower with IC_50_ value 48.7 μM and 103.7 μM, respectively. Compound **CAP 6** showed a selective, strong inhibitory effect in Jurkat cells with IC_50_ 8.5 μM. In other cell lines, the inhibitory effect of **CAP 6** was lower, ranging from 30.5–44.5 μM and 98.8 μM in PANC-1, respectively. All tested compounds exhibited significantly stronger inhibitory effects than caffeic acid itself. Caffeic acid itself showed no significant inhibitory activity in the whole tested cell spectrum. In the same tested concentration range (1–100 µM), the IC_50_ was predictively calculated with concentrations above 300 µM. As reference chemotherapy, we used cisplatin (6.3–35.4 µM). Compared with standard, synthesized caffeic acid derivatives showed close IC_50_, except for response in pancreatic adenocarcinoma cell lines. Higher cytotoxicity of **CAP 8**, **CAP 10**, and **CAP 12** was detected for the cervical cancer cell lines, mammary gland adenocarcinoma cell lines, metastatic melanoma cell lines, and leukemic T cell lymphoma cell lines (Jurkat), where the best response was detected. The comparison of CAPs and CA in leukemic Jurkat cells is presented in Figure 3.

Our newly synthesized compounds are highly specific in their structure. Nevertheless, it seems that prolongation of the alkyl chain in the CAPs enhances its anticancer activity, as it can be expected that an increase in lipophilicity facilitates the penetration of compounds into the cell and subsequently into the mitochondria. For example, decylcaffeic acid inhibits the growth of colorectal cancer cells (HT-29 and HCT-116) through induction of cell cycle arrest at the S phase by cyclin E and cyclin A down-regulation and blockade of the Akt and STAT3 proteins, whereas its activity is approximately 10 times higher in comparison to caffeic acid [27]. Caffeic acid *p*-nitrophenethyl ester and caffeic acid phenethyl ester (CAPE) displayed anticancer effects in colorectal cancer cells (HT-29 and HCT-116) by decreasing cell viability, promoting apoptosis and cell cycle arrest via the P53 pathway and inhibiting cancer cell growth [29]. Higher cytotoxic activity and migration rate inhibition were observed in breast cancer MDA-MB-231 cells for CAPE compared to non-substituted caffeic acid [24,25]. CAPE has also been shown to be cytotoxic in breast cancer MCF-7 cell lines, where it induces cell cycle arrest by upregulating the expression of cycle arrest control genes. In both MCF-7 and MDA-231, CAPE inhibits growth via apoptosis and the cell cycle without much effect on normal mammary cells [30]. CAPE inhibits pancreatic cancer PANC-1-cell cell growth and migration induced by human neutrophil elastase [31].

In the present study, phosphanium salts derived from caffeic acids have exhibited cytotoxic effects comparable to a chemotherapeutical reference compound. This selective antitumor activity should be considered in further investigations or applications.

### 2.2. Antibacterial and Antifungal Activity

The antibacterial activity of caffeic acid and its derivatives was tested against common bacterial species isolated from nosocomial infections, affecting predominantly a predisposed patient weakened in immunoreactivity by various underlying conditions [32]. The selection of species covered Gram-positive bacteria (*Enterococcus faecalis*, methicillin-susceptible and methicillin-resistant strains of *Staphylococcus aureus*) and Gram-negative species (a non-fermenting rod *Pseudomonas aeruginosa*, and fermenting intestinal rods *Escherichia coli*, *Klebsiella pneumoniae*, and *Proteus mirabilis*). The antifungal activity was detected against a strain of the yeast *Candida albicans*. End-point tests were used, and minimal inhibitory and minimal microbicidal activity was detected.

Caffeic acid’s antibacterial activity was not too high (Table 3), probably due to its unsatisfactory ability to cross the biological membranes effectively [33]. The highest antibacterial activity was detected against *Pseudomonas aeruginosa* (MIC/MBC = 3469 μM, corresponding to 625 μg/mL). The caffeic acid MIC/MBC values in the case of the other tested bacteria and the strain of *Candida albicans* exceeded 6900 μM (1200 μg/mL). The results of the obtained caffeic acid antimicrobial activity match the results of former studies with corresponding test conditions [9]. Even if various semi-synthetic caffeic acid derivatives have already been studied, according to available reports, no research on antimicrobial activity has been performed with caffeic acid quaternary phosphanium bromide derivatives until now. While the minimal inhibitory and microbicidal concentrations of caffeic acid reached or even exceeded the upper border defined by Lipinski’s rule-of-five for drug-like compounds [34], derivatization resulted in a strong decrease of MIC, MBC, and MFC values thanks to changes in the physicochemical properties of the molecules. The potential antimicrobial mechanism of action of CAPs could be explained, like the effects of cationic surfactants. Cationic surfactants are well-known antimicrobial agents that affect Gram-positive and Gram-negative bacteria and yeasts. Cationic surfactants applied at alkaline to neutral pH exhibit high affinity to the negatively charged surface of the microbial membrane. Damage to the cell membrane is mediated by binding to phospholipids, which leads to the loss of structural integrity of the cytoplasmic membrane; the penetration of the antibacterial substance into the cell increases and induces the leakage of intercellular components and cell lysis [35,36].

Depending on the tested microorganisms, one- to three-ordered reduction of MIC, MBC and MFC values has been noticed, which means almost 1000-times increased antimicrobial activity against some microbes, as seen in Table 3 against *Enterococcus* and *Candida* (MIC and MBC/MFC decrease, resulting in nearly 1000-fold increased inhibitory and microbicidal activity), and against staphylococci (MBC decrease, resulting in 1000-fold increase in microbicidal activity). In general, the tested caffeic acid derivates were more effective against Gram-positive bacteria and the yeast *Candida albicans*, where the decrease in the inhibitory and microbicidal concentrations was the most prominent. A similar effect was also proved in the study with rosmarinic acid derivates [15].

It is known that the antimicrobial properties of amphiphilic derivatives tend to increase with increasing alkyl chain length till a limit [33,37]. However, this phenomenon was not observed with caffeic acid quaternary phosphanium derivatives. On the contrary, a slight decrease in antimicrobial activity with increasing molecule size (despite increasing lipophilicity) has been noticed in the case of *E. coli* and *K. pneumoniae*, similar to results with *Acanthamoeba* (see Table 4). The molecular size and lipophilicity of caffeic acid derivatives did not affect the inhibitory or microbicidal activity in the case of the other tested microorganisms.

### 2.3. Anti-Acanthamoeba Activity

The anti-*Acanthamoeba* activity was studied against two pathogenic *Acanthamoeba* strains isolated from the corneas of patients with *Acanthamoeba* keratitis (AK) [38]. Although AK is the most frequent *Acanthamoeba* infection, these opportunistic protozoan parasites also cause granulomatous amoebic encephalitis (GAE), cutaneous and disseminated infections, mainly in immunodeficient individuals [39,40]. For all those infections, no uniform and easily manageable treatment has been established to date, and that leads to the continual development of new amoebicidal drugs [13,41].

CA is an important component of plant extracts exhibiting anti-*Acanthamoeba* activity, e.g., extracts from *Eryngium* spp. [42] and *Olea europaea* [43]. In our experiments, we detected a relatively weak cytotoxic effect of CA with values of EC_50_ ≥ 3005.00 µM after 24 h (Table 4). This is in accordance with results published by Sifaoui et al. (2017), who detected 50% growth inhibition of non-pathogenic Neff’s strain of *Acanthamoeba castellanii* caused by CA with a value of IC_50_ > 100 µg/mL after 96 h of incubation [44]. However, our results show that the derivatives of CA are distinctly more effective. The tested phosphanium salts exhibited considerably higher inhibitory activity than CA except for **CAP 12**, which showed a slightly lower amoebicidal effect on the *A. quina* strain. The highest amoebicidal effect clearly demonstrated the compound **CAP 6**. Its EC_50_ value was more than two times lower than the EC_50_ of **CAP 8**, more than six times lower than the EC_50_ of **CAP 10**, and more than 75 times lower than the EC_50_ of **CAP 12** for *A. quina*. In the case of *A. lugdunensis*, the EC_50_ of **CAP 6** was distinctly lower than the EC_50_ of **CAP 8**, more than 11 times lower than the EC_50_ of **CAP 10**, and more than 42 times lower than the EC_50_ of **CAP 12**. The detected decrease in the amoebicidal activity of the compounds with the increasing number of carbon atoms in the connecting chain between the ester group and the phosphanium cation correlates with an increase in the lipophilicity of the compounds. Similarly, the lower anti-*Acanthamoeba* activity of the compounds with higher lipophilicity was also detected in several previous studies [39,45]. Experiments also showed that the susceptibility of *Acanthamoeba* isolates to tested compounds differed markedly. The *A. lugdunensis* strain was less susceptible to CA, **CAP 6**, **CAP 8**, and **CAP 10** than the *A. quina* strain. The *A. quina* strain was less susceptible than the *A. lugdunensis* strain only to the compound **CAP 12**.

Our compounds are structurally similar to phosphanium salts of rosmarinic acid, which showed only moderate anti-*Acanthamoeba* activity [15]. However, in the present study, **CAP 6** demonstrated more than 15 times higher inhibitory activity in comparison to those compounds. The amoebicidal activity of the derivatives of CA has not been studied to date, and the present results clearly show that the derivatization of naturally occurring plant metabolites may bring very interesting results of anti-*Acanthamoeba* activity.

Based on our results, the activity of CAPs on some tumor cells, most of the tested bacteria, and yeast was found to be independent of the length of the alkyl chain connecting the CA and the phosphanium cation. Against most of the tested cancer cell lines, the cytotoxic activity increased with increasing lipophilicity, while against acanthamoeba and some of the tested Gram-negative bacteria, this dependence was found to be inverse; the activity of compounds slightly decreased with increasing molecular size and lipophilicity.

By adding a phosphanium chain to caffeic acid, we prepared four substances with surfactant properties. Surfactants as pharmaceuticals are well-known for topical application, although some surfactants within acceptable dose limits are considered safe for human consumption and could be used in oral pharmaceutical products [46]. According to our results, the prepared caffeic acid phosphanium derivatives exhibited a high cytotoxic effect against the control cell line. This fact limits their oral administration and determines them as topical agents for further research.

## 3. Materials and Methods

A caffeic acid was purchased by Sigma Aldrich (Sigma-Aldrich, St. Louis, MO, USA). All other chemicals used in the synthesis were obtained from commercial supplier (Merck, Darmstadt, Germany) and were of p.a. purity. ^1^H-, ^13^C-, and ^31^P-NMR spectra were measured on a MERCURY plus spectrometer (Varian, Palo Alto, CA, USA), working at frequencies of 300, 75, and 121.5 MHz, respectively. ^13^C- and ^31^P-NMR spectra were coupled against protons. The spectra were measured in CDCl_3_. The chemical shifts were referenced with respect to an internal TMS (δ ^1^H = 0, δ ^13^C = 0) or 85% H_3_PO_4_ (δ ^31^P = 0 for ∈ ^31^P = 40.4807420 MHz) signal.

### 3.1. Synthesis

#### 3.1.1. Acetylation of Caffeic Acid

Caffeic acid (15 g) was dissolved in a 250 mL flask in anhydrous pyridine (50 mL), and the distillation flask was closed with a septum. Acetic anhydride (44 mL) was added with a syringe. The reaction mixture was stirred for 24 h at room temperature in a nitrogen atmosphere. Then, the crude product was poured into a separatory funnel (500 mL), dichloromethane (200 mL) was added, and the solution was extracted with 3 molar HCl (3 × 50 mL). Subsequently, the dichloromethane solution was washed with saturated NaCl solution (50 mL) and dried with anhydrous MgSO_4_. The drying agent was removed by filtration, and the crude product was further purified by crystallization from methanol (50 mL). Acetylated caffeic acid was dried under a vacuum.

(*2E*)-3-(3,4-diacetyloxyphenyl)prop-2-enoic acid. Yield: 88.9%; ^1^H NMR (300 MHz, CDCl_3_) δ: 2.27 (s, 3H), 2.32 (s, 3H), 6.39 (d, *J* = 16.2 Hz, 1H) 7.23–7.45 (m, 3H), 7.72 (d, *J* = 15.9 Hz, 1H); ^13^C NMR (75 MHz, CDCl_3_) δ: 20.6, 20.7, 118.4, 123.0, 124.0, 126.7, 132.9, 142.5, 143.9, 145.0, 168.0, 168.1, 171.6. The NMR spectra are in good agreement with the literature [47].

#### 3.1.2. General Procedure for Esterification of (*2E*)-3-(3,4-Diacetyloxyphenyl)prop-2-enoic Acid

(2*E*)-3,4-Diacetoxyphenyl)prop-2-enoic acid (3.5 g, 0.013 mol), α,ω-dibromoalkane (0.052 mol), potassium carbonate (7.41 g, 0.052 mol) was suspended in anhydrous dimethylformamide (35 mL). The reaction mixture was stirred at room temperature for 6 h. Esterification was carried out under a nitrogen atmosphere. After 6 h, the contents of the flask were poured into a separatory funnel, water (200 mL) was added, and the aqueous layer was extracted with ethyl acetate (4 × 50 mL). The organic layers were combined and subsequently washed with water (3 × 50 mL) and brine (50 mL). The ethyl acetate solution was dried over anhydrous Na_2_SO_4_, then the drying agent was filtered off, and the ethyl acetate was evaporated in a vacuum. The product was purified by chromatography using silica gel with the liquid phase containing petroleum ether:ethyl acetate (15:1 → 1:1).

6-bromohexyl-(2*E*)-3-[3,4-di(acetoxy)phenyl]prop-2-enoate. Yield: 60.8%; ^1^H NMR (300 MHz, CDCl_3_, TMS) δ: 1.39–1.52 (m, 4H), 1.65–1.78 (m, 2H), 1.85–1.92 (m, 2H), 2.30 (s, 3H), 2.31 (s, 3H), 3.42 (t, *J* = 6.9 Hz, 2H), 4.20 (t, *J* = 6.9 Hz, 2H), 6.38 (d, *J* = 15.9 Hz, 1H), 7.20–7.28 (m, 1H), 7.35-7.42 (m, 2H), 7.61 (d, *J* = 15.9 Hz, 1H); ^13^C NMR: (75 MHz, CDCl_3_, TMS) δ: 20.6(1), 20.6(5), 25.2, 27.8, 28.5, 32.6, 33.7, 64.6, 119.4, 122.7, 123.9, 126.4, 133.3, 142.4, 142.7, 143.5, 166.6, 168.0, 168.1.8-bromooctyl-(2*E*)-3-[3,4-di(acetoxy)phenyl]prop-2-enoate. Yield: 48.8%; ^1^H NMR (300 MHz, CDCl_3_) δ: 1.35–1.45 (m, 8H), 1.67–1.70 (m, 2H), 1.84–1.89 (m, 2H), 2.31 (s, 3H), 2.32 (s, 3H), 3.41 (t, *J* = 6.9 Hz, 2H), 4.20 (t, *J* = 6.6 Hz, 2H), 6.38 (d, *J* = 15.9 Hz, 1H), 7.21–7.42 (m, 3H), 7.61 (d, *J* = 15.9 Hz, 1H); ^13^C NMR (75 MHz, CDCl_3_) δ: 20.6(1), 20.6(4), 25.9, 28.1, 28.6, 29.1, 32.8, 34.0, 64.8, 119.5, 122.7, 123.9, 126.4, 133.4, 142.4, 142.6, 143.4, 166.7, 168.0, 168.1.10-bromodecyl-(2*E*)-3-[3,4-di(acetoxy)phenyl]prop-2-enoate. Yield: 54.8%; ^1^H NMR (300 MHz, CDCl_3_) δ: 1.22–1.45 (m, 12H), 1.62–1.75 (m, 2H), 1.81–1.92 (m, 2H), 2.30 (s, 3H), 2.31 (s, 3H), 3.41, (t, *J* = 6.9 Hz, 2H), 4.19 (t, *J* = 6.9 Hz, 2H), 6.38 (d, *J* = 15.9 Hz, 1H), 7.21–7.30 (m, 1H), 7.35–7.42 (m, 2H), 7.61 (d, *J* = 15.9 Hz, 1H); ^13^C NMR: (CDCl_3_, TMS) δ: 20.6, 20.7, 25.9, 28.1, 28.7, 28.7, 29.2, 29.3, 29.4, 32.8, 34.0, 64.8, 119.5, 122.7, 123.9, 126.4, 133.4, 142.4, 142.6, 143.4, 166.7, 168.0, 168.1.12-bromododecyl-(2*E*)-3-[3,4-di(acetoxy)phenyl]prop-2-enoate. Yield: 41.8%; ^1^H NMR (300 MHz, CDCl_3_) δ: 1.26–1.56 (m, 16H), 1.67–1.69 (m, 2H), 1.83–1.88 (m, 2H), 2.30 (s, 3H), 2.31 (s, 3H), 3.41 (t, *J* = 6.9 Hz, 2H), 4.19 (t, *J* = 6.9 Hz, 2H), 6.38 (d, *J* = 15.9 Hz, 1H), 7.21–7.41 (m, 3H), 7.60 (d, *J* = 15.9 Hz, 1H); ^13^C NMR (75 MHz, CDCl_3_) δ: 20.6, 25.9, 28.2, 28.7, 28.8, 29.3, 29.4(8), 29.5, 32.8, 34.1, 64.9, 119.5, 122.7, 123.9, 126.4, 133.4, 142.5, 143.4, 166.7, 168.0, 168.1.

#### 3.1.3. General Procedure for Quaternization of ω-Bromoalkyl (2*E*)-3-[3,4-Di(acetoxy)phenyl]prop-2-enoates with Cfuane

ω-Bromoalkyl (2*E*)-3-[3,4-di(acetoxy)phenyl]prop-2-enoate (5 mmol) and triphenyphospane (1.57 g, 6 mmol) were dissolved in acetonitrile 20 mL. The reaction mixture was allowed to stir for 24 h at 80 °C. The reaction was performed in a nitrogen atmosphere. After completion of the reaction, the solvent was evaporated, and the crude product was purified by column chromatography using silica gel with the liquid phase containing petroleum ether:chloroform (1:1) → chloroform → chloroform:methanol (15:1).

*P*,*P*,*P*-triphenyl-6-{(2*E*)-3-[3,4-(diacetyloxy)penyl]prop-2-enoyloxy}hexane-1-phosphanium bromide **CAP 6**. Yield: 73.6%; ^1^H NMR (300 MHz, CDCl_3_) δ: 1.35–1.45 (m, 2H), 1.58–1.75 (m, 6H), 2.30 (s, 3H), 2.31 (s, 3H), 3.78–3.89 (m, 2H), 4.13 (t, *J* = 6.6 Hz, 2H), 6.35 (d, *J* = 15.9 Hz, 1H), 7.28–7.36 (m, 1H), 7.36–7.43 (m, 2H), 7.62 (d, *J* = 15.9 Hz, 1H), 7.69–7.90 (m, 15H); ^13^C NMR: (CDCl_3_, TMS) δ: 20.7, 22.4, 22.6, 22.9 (d, *J* = 30.4 Hz), 25.7, 28.3, 30.0 (d, *J* = 16.1 Hz), 64.5, 118.4 (d, *J* = 85.1 Hz), 119.4 122.7, 123.9, 126.4, 130.5 (d, *J* = 12.5 Hz), 133.3, 133.7 (d, *J* = 9.9 Hz), 135.0 (d, *J* = 3.0 Hz), 142.4, 142.7, 143.5, 166.6, 168.0, 168.1; ^31^P NMR: (CDCl_3_, TMS) δ: 24.4.*P*,*P*,*P*-triphenyl-8-{(2*E*)-3-[3,4-(diacetyloxy)penyl]prop-2-enoyloxy}okctane-1-phosphanium bromide **CAP 8**. Yield: 67.0%; ^1^H NMR (300 MHz, CDCl_3_) δ: 1.26–1.29 (m, 6H), 1.63–1.81 (m, 6H), 2.17 (s, 3H), 2.30 (s, 3H), 3.78 (t, *J* = 5.4 Hz, 2H), 4.14 (t, *J* = 6.6 Hz, 2H), 6.37 (d, *J* = 15.9 Hz, 1H), 7.20–7.42 (m, 3H), 7.59 (d, *J* = 15.9 Hz, 1H) 7.67–7.88 (m, 15H); ^13^C NMR (75 MHz, CDCl_3_) δ: 20.7, 22.4, 22.6, 22.9 (d, *J* = 30.1 Hz), 25.8, 28.6, 28.8, 29.1, 30.2 (d, *J* = 15.8 Hz), 64.7, 118.4 (d, *J* = 85.2 Hz), 119.5, 122.7, 123.9, 126.4, 130.5 (d, *J* = 12.5 Hz), 133.3, 133.7 (d, *J* = 9.9 Hz), 135.0 (d, *J* = 2.9 Hz), 142.4, 142.6, 143.4, 166.7, 168.0, 168.1; ^31^P NMR (121,47 MHz, CDCl_3_) δ: 24.4.*P*,*P*,*P*-triphenyl-10-{(2*E*)-3-[3,4-(diacetyloxy)penyl]prop-2-enoyloxy}decane-1-phosphanium bromide **CAP 10**. Yield: 72.5%; ^1^H NMR (300 MHz, CDCl_3_) δ: 1.21–1.42 (m, 12H), 1.55–1.82 (m, 4H), 2.30 (s, 3H), 2.31 (s, 3H), 3.75–3.83 (m, 2H), 4.17 (t, *J* = 6.9 Hz, 2H), 6.38 (d, *J* = 16.2 Hz, 1H), 7.21–7.28 (m, 1H), 7.35–7.42 (m, 2H), 7.61 (d, *J* = 16.2 Hz, 1H), 7.65–7.90 (m, 15H); ^13^C NMR: (CDCl_3_, TMS) δ: 206, 22.5, 22.6, 22.9 (d, *J* = 29.6 Hz), 25.9, 28.6, 29.0(7), 29.1, 29.2, 29.3, 30.4 (d, *J* = 15.6 Hz), 64.8, 118.4 (d, *J* = 85.3 Hz), 119.3 122.7, 123.9, 126.4, 130.5 (d, *J* = 12.5 Hz), 133.3, 133.7 (d, *J* = 9.8 Hz), 134.9 (d, *J* = 3.0 Hz), 142.4, 142.6, 143.4, 166.7, 168.0, 168.1; ^31^P NMR: (CDCl_3_, TMS) δ: 24.5.*P*,*P*,*P*-triphenyl-12-{(2*E*)-3-[3,4-(diacetyloxy)penyl]prop-2-enoyloxy}dodecane-1-phosphanium bromide **CAP 12**. Yield: 65.5%; ^1^H NMR (400 MHz, CDCl_3_) δ: 1.20–1.37 (m, 16H), 1.62–1.70 (m, 4H), 2.30 (s, 3H), 2.31 (s, 3H), 3.80–3.81 (m, 2H), 4.18 (t, *J* = 6.4 Hz, 2H), 6.38 (d, *J* = 15.9 Hz, 1H), 7.22 (d, *J* = 8.4 Hz, 1H), 7.36 (d, *J* = 2.0 Hz, 1H), 7.40 (d, *J* = 8.4 Hz, *J* = 2.0 Hz, 1H), 7.61 (d, *J* = 15.9 Hz, 1H), 7.68–7.88 (m, 15H); ^13^C NMR (100 MHz, CDCl_3_) δ: 20.6, 20.7, 22.5, 22.9 (d, *J* = 30.1 Hz), 25.9, 28.7, 29.2, 29.2, 29.2, 29.4, 30,4 (d, *J* = 15.6 Hz), 64.8, 118.5 (d, *J* = 85.7 Hz), 119.5, 122.7, 123.9, 126.3, 130.4 (d, *J* = 12.2 Hz), 133.3, 133.7 (d, *J* = 9.9 Hz), 134.9 (d, *J* = 3.1 Hz), 142.4, 142.6, 143.4, 166.7, 168.0, 168.1; ^31^P NMR (162 MHz, CDCl_3_) δ: 24.5;

The NMR spectra obtained in this study are shown in the Appendix A.

#### 3.1.4. Equilibrium Surface Tension Measurements

The equilibrium surface tension measurements were carried out according to the procedure described previously [48]. The Wilhelmy plate technique was used for the determination of the solvent surface tension values. Measurements were performed with a Krüss 100 MK2 tensiometer. All samples were prepared by dissolving amphiphilic compounds in deionized water. The stock solutions were prepared in a volumetric flask. The measurements were performed at 25 ± 0.1 °C. The measurements of equilibrium surface tension were taken repeatedly. The values were recorded every 360 s. The measurement was stopped if the difference between the values of two successive measurements was less than 5 × 10^−5^ N·m^−1^. The breakpoint of the linear parts of the surface tension vs. log c curve was used for the determination of the critical micelle concentration (cmc) and surface tension at the cmc (γ_cmc_). The adsorbed amount of surfactant Γ_cmc_ was also determined from surface tension data. The value was calculated using the Gibbs adsorption isotherm:$$\Gamma_{cmc}=\frac{{\left[\frac{d\gamma }{dlogc}\right]}_{T}}{\left(2.303iRT\right)}$$
where *γ* represents the surface tension (mN/m), *c* is the concentration of surfactant, *i* represents the prefactor (CAP have *i* = 2), *R* is the gas constant, and *T* represents the absolute temperature. The slope below the *cmc* in the surface tension vs. log *c* plots was used for the determination of the surface excess. The values of area per head group at the water/air interface (*A_cmc_*) were obtained from the following equation:$$A_{cmc}=\frac{10^{18}}{N_{A}\Gamma_{cmc}}$$
where *N_A_* represents the Avogadros constant.

#### 3.1.5. HPLC-DAD Analysis of CAPs

The purity of synthesized compounds was evaluated by HPLC-DAD. The HPLC-DAD analyses were performed using an HPLC system (Sykam, Eresing, Germany) equipped with a pump (S1125), an autosampler (S5250), a column oven (S4120), PDA detector (S3345), and Clarity Software Version 8.8.1.16. We used HPLC column 250 mm × 4.6 mm Ultisil^®^ AQ-C18 3 μm (Welch, West Haven, CT, USA) at a constant column temperature of 25 °C. The injection volume of all experiments was 20 mL. The analysis was performed according to Stojanovic et al. (2008) [49]. The DAD wavelength range was chosen from 200 to 400 nm, with a data acquisition rate of 2 Hz and the absorption signal at 254 nm. HPLC experiments were performed using 0.05% TFA (trifluoroacetic acid 98%, Sigma-Aldrich, St. Louis, MO, USA) in HPLC-grade water (Merck, Darmstadt, Germany) as eluent A and HPLC-grade acetonitrile (Merck, Darmstadt, Germany) with 0.1% TFA as eluent B. The final optimized method made use of the following step-gradient elution mode: 40% B from 0 to 10 min, then up to 80% B in 0.5 min and held constant at 80% B from 10.5 to 25 min, followed by a decrease to 40% B in 0.5 min and then re-equilibration from 25.5 to 30 min at 40% B. The mobile phase flow rate was 1.0 mL/min, except for the re-equilibration period, for which it was set to 1.5 mL/min. Each sample was dissolved in the initial mobile phase (60% H_2_O + 0.05% TFA/40% ACN + 0.05% TFA) to a final concentration of 0.1 mg/mL. Three independent measurements were performed for precision and accuracy.

Chromatograms are available in Appendix A.

### 3.2. Biological Activities

The compounds were dissolved in dimethyl sulfoxide (DMSO, Sigma-Aldrich, St. Louis, MO, USA). The final concentration of DMSO in the culture medium was <0.2% and exhibited no cytotoxicity.

#### 3.2.1. MTS Cell Proliferation/Viability Assay

The human cancer cell line HCT116 (human colorectal adenocarcinoma), HeLa (human cervical adenocarcinoma), Jurkat (human leukemic T cell lymphoma), and MDA-MB-231 (human mammary gland adenocarcinoma, triple-negative) were cultured in RPMI 1640 medium (Biosera, Kansas City, MO, USA). MCF-7 (human breast adenocarcinoma), A2058 (human metastatic melanoma), PANC-1 (human pancreatic adenocarcinoma), and NIH 3T3 (murine fibroblasts) cell lines were maintained in a growth medium consisting of high glucose Dulbecco’s Modified Eagle Medium with sodium pyruvate (Biosera, Kansas City, MO, USA). MCF-10A cell line (human mammary gland epithelial cells) was maintained in a growth medium consisting of high glucose DMEM F12 Medium (Biosera, Kansas City, MO, USA) + Suppl. (insulin, EGF-epithelial growth factor, HC-hydrocortisone, choleratoxin all Sigma, Steinheim, Germany). The growth medium was supplemented with a 10% fetal bovine serum, penicillin (100 IU/mL), and streptomycin (100 μg/mL) (all Invitrogen, Carlsbad, CA, USA), and cells were maintained in an atmosphere containing 5% CO_2_ in humidified air at 37 °C. Cell viability, estimated by trypan exclusion, was greater than 95% before each experiment.

The metabolic activity colorimetric assay (MTS) was used to determine the effects of CA (c = 10–300 µM) and compounds **CAP 6**, **CAP 8**, **CAP 10**, and **CAP 12** (c = 1–100 µM) on the metabolic activity of several cell lines. After 72 h of incubation, 10 µL of MTS (Promega, Madison, WI, USA) was added to each well according to the CellTiter 96^®^ AQueous One Solution Cell Proliferation Assay protocol. After a minimum of 1 h incubation, the absorbance was measured at 490 nm using the automated Cytation^TM^ 3 Cell Imaging Multi-Mode Reader (Biotek, Winooski, VT, USA). The absorbance of the control wells was taken as 1.0 (100%), and the results were expressed as a fold of the control. All experiments were performed in triplicate. Ideal predictive IC_50_ values were calculated from MTS analyses-based concentration concentration-dependent trends.

#### 3.2.2. Antibacterial and Antifungal Activity Testing

The antimicrobial activity of caffeic acid and its semi-synthetic derivates was tested against Gram-positive (*Staphylococcus aureus* CCM 4750, methicillin-resistant; *Staphylococcus aureus* CCM 4223, methicillin-susceptible; *Enterococcus faecalis* CCM 4224) and Gram-negative bacteria (*Pseudomonas aeruginosa* CCM 3955; *Escherichia coli* CCM 3954; *Klebsiella pneumoniae* CCM 4415; *Proteus mirabilis* CCM 7188), and one strain of a yeast *Candida albicans* (CCM 8267). Microbial strains were purchased from the Czech Collection of Microorganisms, Brno, Czech Republic, and preserved in aliquots in cryoprotective media at −20 °C until the testing. Revival of cryoprotected bacteria took place by overnight cultivation on blood agar and an additional passage on blood agar. Minimal inhibitory concentrations (MIC) against bacteria were detected by broth microdilution test in sterile 96-well U-shaped microtiter plates (Roll s.a.s., Piove di Sacco, Italy) according to the EUCAST recommendations [50]. The test was performed in 100 μL volumes in microtiter plate wells with a final bacterial concentration of 5 × 10^5^ CFU·mL^−1^. Antifungal activity testing was performed according to the EUCAST-recommended method for the determination of broth dilution minimum inhibitory concentrations of antifungal agents for yeasts [51], with some modifications. The final yeast concentration in the test wells was 10^4^ CFU·mL^−1^. The minimal bactericidal/fungicidal concentrations (MBC/MFC) were detected by point-inoculation of 5 μL aliquots from test wells without visible growth on solid agar media free of antimicrobials. Mueller–Hinton broth and agar were used for bacteria, and Sabouraud dextrose broth and agar for fungi (OXOID, Basingstoke, Hampshire, UK). The tested agents were dissolved in 50% ethanol, and serial dilutions were prepared in the test medium to obtain final concentrations ranging from 5000 to 5 μg·mL^−1^, which corresponds to 27,753—54 μM for caffeic acid, 7251—14 μM for **CAP 6**, 6967—14 μM for **CAP 8**, 6705—13 μM for **CAP 10** and 6462—13 μM for **CAP 12**. Wells with media free of bacteria served as sterility control and wells with bacteria without antimicrobial agents were used as growth control. The MIC was determined to be the lowest concentration of the antimicrobial agent inhibiting the growth of the tested microorganism, and the MBC/MFC was determined to be the lowest concentration, inactivating 99.9% of microbial inoculum. All tests were performed in three independent runs to exclude possible excessive values due to permissible measurement error.

#### 3.2.3. Amoebicidal Activity Assay

The activity against two clinical isolates of free-living amoebae, *Acanthamoeba lugdunensis* (strain AcaVNAK02) and *Acanthamoeba quina* (strain AcaVNAK03), both belonging to the T4 genotype, was tested in vitro as previously described [38]. The trophozoites were cultivated in 25 cm^2^ tissue culture flasks under axenic conditions in peptone-yeast extract-glucose medium (PYG). The cultivation was carried out under aerobic conditions at laboratory temperature. For the experiments, a fresh 72 h culture was used. Acanthamoebae adherent to flasks represented the trophozoite form and was used in the subsequent assay. Experiments were carried out in 96-well microtiter plates under sterile conditions at 37 °C. Each well was seeded with 100 µL (2 × 10^5^ cells mL^−1^) of a trophozoite suspension. Then, 100 µL of a freshly prepared medium containing a tested compound at 6 different concentrations was added to all wells, except for the untreated control wells that received 100 µL of the pure medium. Each compound was tested at final concentrations of 500, 250, 125, 62.5, 31.25, and 15.6 µM. The reduction of trophozoites was recorded after 24 h by counting the surviving cells in a Bürker–Türk hemocytometer. The growth control after 24 h (amoebae in inoculum without treatment): *A. lugdunensis* (AcaVNAK02): 2.5 × 10^5^ cells mL^−1^, *A. quina* (AcaVNAK03): 2.1 × 10^5^cells/mL comparing to initial inoculum 1 × 10^5^ cells mL^−1^. The viability of trophozoites was determined by trypan blue exclusion. The EC_50_ (effective concentration of tested compound that reduces the survival of amoebae by 50%) values were calculated by linear regression analysis using Microsoft Office Excel 2016 (Microsoft Corporation, Redmond, WA, USA) software. All experiments were performed in quadruplicate for each concentration.

#### 3.2.4. Statistical Analyses

Results are expressed as mean ± standard deviation (SD). Statistical analyses of the data were performed using standard procedures *t*-test. Differences were considered significant when *p* < 0.05. Throughout this paper, * indicates *p* < 0.05, ** *p* < 0.01, *** *p* < 0.001 versus untreated cells.

## 4. Conclusions

Plants produce many health-promoting compounds that are used in therapy or prevention. One of the well-known biologically active secondary metabolites with ubiquitous occurrence in the plant kingdom is caffeic acid, which has various valuable biological effects, such as antioxidant, anti-inflammatory, or antimicrobial. Our study answers whether it is possible to improve the pharmacological effects of caffeic acid by adding a phosphanium chain to its structure. Through this chemical intervention, we prepared four salts from CA with surfactant-like properties and improved their biological profile. Antitumor activity was significantly increased, comparable to standard cisplatin for selective cancer cell lines. The high potential of the substances was noted in the case of antimicrobial action, especially against Gram-positive bacteria and *Candida albicans*, and even against protozoa *Acanthamoeba lugdunensis* and *A. quina*. In conclusion, the high cytotoxicity of caffeic acid phosphanium salts limits oral administration; however, they can be considered potential topical agents for further research or might inspire the design of related compounds.

## Figures and Tables

**Figure 1 ijms-25-01200-f001:**
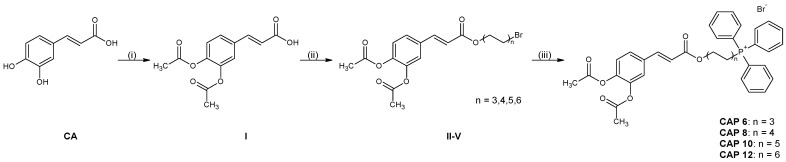
Preparation of CAPs: (i) acetic anhydride, pyridine; (ii) K_2_CO_3_, α,ω-dibromoalkane, DMF; (iii) triphenylphosphane, acetonitrile.

**Figure 2 ijms-25-01200-f002:**
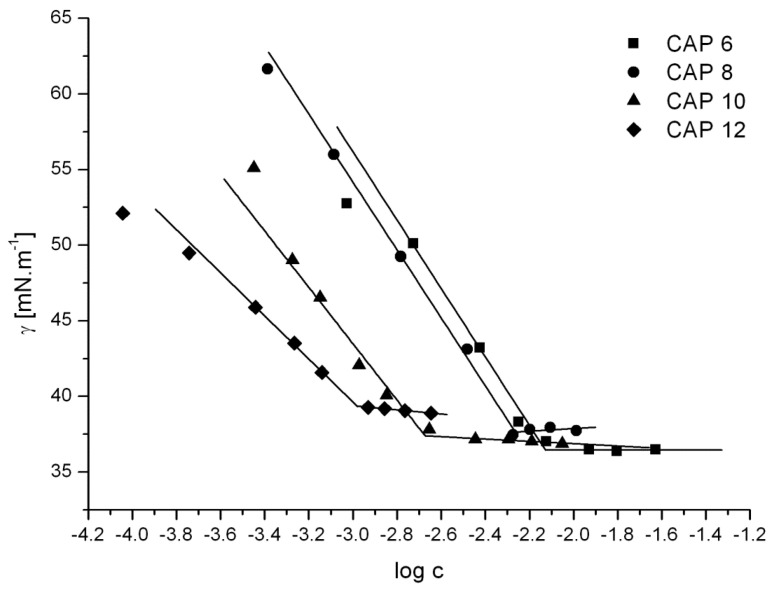
Plots of surface tension vs. log concentration of CAPs.

**Figure 3 ijms-25-01200-f003:**
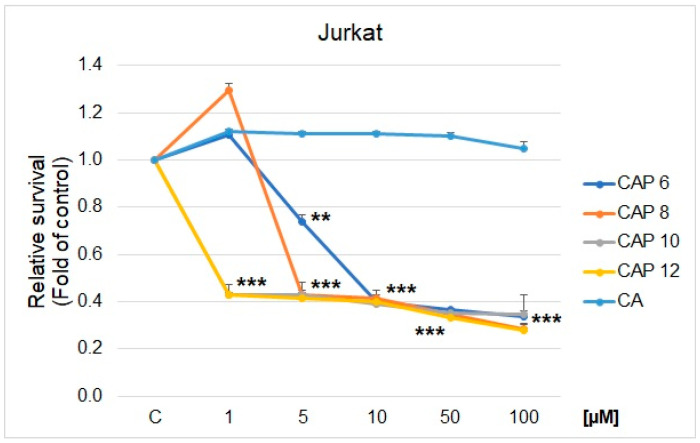
MTS analyses of CAPs derivatives in Jurkat cells after 72 h treatment. Significant difference: ** *p* > 0.01, *** *p* > 0.001.

**Figure 4 ijms-25-01200-f004:**
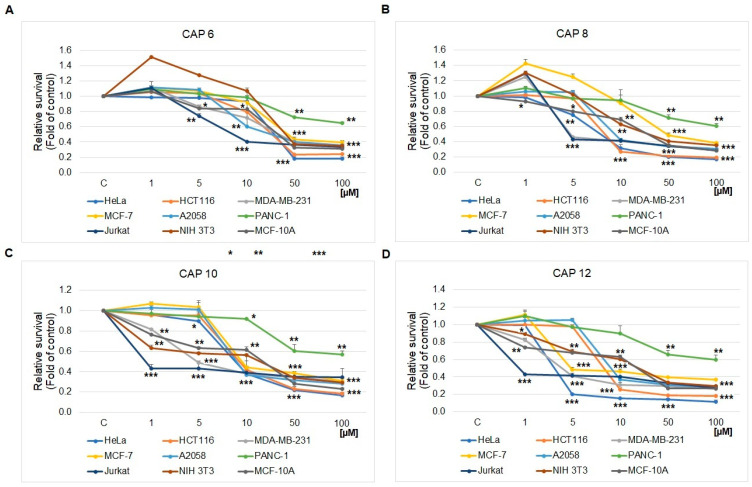
MTS analyses of **CAP 6** (**A**), **CAP 8** (**B**), **CAP 10** (**C**), and **CAP 12** (**D**) derivatives in several cancerous and non-cancerous cell lines after 72 h treatment. Significant difference: * *p* > 0.05, ** *p* > 0.01, *** *p* > 0.001.

**Table 1 ijms-25-01200-t001:** Surface tension data of new caffeic acid derivatives.

Compound	cmc [mol·dm^−3^]	γ_cmc_ [mN·m^−1^]	A_cmc_ [nm^2^]
**CAP 6**	7.5 × 10^−3^	36.4	0.84
**CAP 8**	5.2 × 10^−3^	37.7	0.82
**CAP 10**	2.2 × 10^−3^	37.1	1.02
**CAP 12**	1.0 × 10^−3^	39.4	1.33

**Table 2 ijms-25-01200-t002:** Cytotoxic activity of caffeic acid and its new derivatives on different cells proliferation/viability.

Compound	Cancer Cell Lines/IC_50_ (μM)							Control Cell Lines/IC_50_ (μM)	
	HeLa	HCT116	MDA-MB-231	MCF-7	A2058	PANC-1	Jurkat	NIH 3T3	MCF-10A
**CAP 6**	34.0 ± 1.8	31.0 ± 0.4	37.0 ± 0.3	44.5 ± 0.1	30.5 ± 1.5	98.8 ± 2.4	8.5 ± 0.2	44.4 ± 2.6	53.7 ± 0.2
**CAP 8**	7.1 ± 0.3	8.3 ± 0.6	4.8 ± 0.5	48.7 ± 0.1	9.4 ± 0.3	103.2 ± 1.3	6.9 ± 0.3	38.2 ± 1.8	42.3 ± 1.1
**CAP 10**	8.0 ± 0.7	9.2 ± 0.3	4.9 ± 0.5	9.5 ± 0.2	9.0 ± 0.5	91.2 ± 2.6	0.9 ± 0.4	28.0 ± 1.3	31.1 ± 2.3
**CAP 12**	3.3 ± 1.1	8.3 ± 0.6	3.5 ± 0.6	4.9 ± 0.7	9.0 ± 0.5	110.7 ± 3.2	0.9 ± 0.2	32.5 ± 0.9	33.7 ± 1.5
CA	>300	>300	>300	>300	>300	>300	>300	>300	>300
CisPt [28]	35.4	7.4	7.1	29.7	21.5	16.5	6.3	40.6	25.9

Control cells: NIH 3T3 (murine fibroblasts) cell lines, MCF-10A cell line (human mammary gland epithelial cells). Cancer cell lines: HeLa (human cervical adenocarcinoma), HCT116 (human colorectal adenocarcinoma), MDA-MB-231 (human mammary gland adenocarcinoma), MCF-7 (human breast adenocarcinoma), A2058 (human metastatic melanoma), PANC-1 (human pancreatic adenocarcinoma), Jurkat (human leukemic T cell lymphoma); Standard: CisPt (cisplatin).

**Table 3 ijms-25-01200-t003:** Antimicrobial activity of caffeic acid and its new derivatives.

**Compound**	**Microbial strains/MIC (μM)**							
	*Staphylococcus aureus* (MR) CCM 4750	*S. aureus* (MS) CCM 4223	*Enterococcus faecalis* CCM 4224	*Pseudomonas aeruginosa* CCM 3955	*Escherichia coli * CCM 3954	*Klebsiella pneumoniae* CCM 4415	*Proteus mirabilis * CCM 7188	*Candida albicans* CCM 8267
CA	6938	6938	13877	3469	6938	6938	6938	13,877
**CAP 6**	14	14	57	453	227	28	453	57
**CAP 8**	27	14	27	871	435	27	218	27
**CAP 10**	13	13	26	838	419	105	210	13
**CAP 12**	25	50	25	404	808	404	404	25
**Compound**	**Microbial strains/MBC or MFC (μM)**							
	*Staphylococcus aureus* (MR) CCM 4750	*S. aureus* (MS) CCM 4223	*Enterococcus faecalis* CCM 4224	*Pseudomonas aeruginosa* CCM 3955	*Escherichia coli * CCM 3954	*Klebsiella pneumoniae* CCM 4415	*Proteus mirabilis * CCM 7188	*Candida albicans * CCM 8267
CA	13,877	13,877	27,753	3469	13,877	6938	6938	13,877
**CAP 6**	14	14	57	453	227	28	453	57
**CAP 8**	27	14	27	871	435	27	435	27
**CAP 10**	26	13	26	676	419	105	419	13
**CAP 12**	25	50	25	808	808	404	404	25

CA—caffeic acid; **CAP 6**–**12**—caffeic acid derivatives; MIC—minimal inhibitory concentration; MBC—minimal bactericidal concentration; MFC—minimal fungicidal concentration; MR—methicillin-resistant; MS—methicillin-susceptible.

**Table 4 ijms-25-01200-t004:** The values of EC_50_ (µM) of caffeic acid and its derivates on *Acanthamoeba* trophozoites after 24 h of incubation. All data are mean values ± standard deviation of four independent experiments.

Compound	*A. lugdunensis* (AcaVNAK02)	*A. quina* (AcaVNAK03)
CA	>10^5^	3005.0 ± 1514.3
**CAP 6**	74.2 ± 18.4	44.1 ± 9.2
**CAP 8**	99.8 ± 23.7	94.1 ± 14.5
**CAP 10**	833.8 ± 303.2	291.1 ± 36.8
**CAP 12**	3124.8 ± 941.0	3323.9 ± 2055.7

## Data Availability

The data supporting the results reported in the presented manuscript are available at the Department of Chemical Theory of Drugs (Miloš Lukáč, lukac@fpharm.uniba.sk) and the Department of Pharmacognosy and Botany, Faculty of Pharmacy Comenius University Bratislava, Slovakia (Silvia Bittner Fialová, fialova@fpharm.uniba.sk).

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
