# Peer review of "Caffeic Acid Phosphanium Derivatives: Potential Selective Antitumor, Antimicrobial and Antiprotozoal Agents"

_ijms, 2024, doi:10.3390/ijms25021200_

Round 1

Reviewer 1 Report

Comments and Suggestions for Authors

The manuscript "Caffeic acid after derivatization revealed selective anticancer, strong antimicrobial, and promising anti-Acanthamoeba activity" by M. Lukáč is devoted to the caffeic acid derivatization and bioactivity testing. The authors describe a few novel phosphonium salts derived from caffeic acid in 3 steps. The main part of the presented material is a results of different assays (amoebocidal, antibacterial, antifungal, cytotoxic effects and surface activity).

1. The authors use underived caffeic acid as a control in all assays. I guess it's not correct: to evaluate the effect and correctness of experimental data we should see the real positive controls, specific for each type of testing. Cytotoxic agent (like doxorubicin), antibacterial, antifungal etc. data should be added to make the conclusions about the compounds' activity. As a result, the discussion section should be extensively rewritten.

2. It's a very specific type of chemical modification - introduction of phosphonium group. Usually it has an aim to redirect the compounds into the mitochondria of eucariotic cells or affect the procariotic cells in similar way (membrane integrity and potential). Here I had not find any explanations why the authors did the presented transformations and what was the aim.

Based on these points I recommend to reject the paper in the current form.

Comments on the Quality of English Language

The section 4. Materials and Methods should be carefully checked and improved.

For example,

line 296: MgSO4 - use subscript for 4;

line 299: 1H NMR - use superscript for 1;

line 301: 13C NMR (75 MHz, CDCl3);

line 306: 7.41g (0.053 mol) potassium carbonate (7.41 g, 0.052 mol);

lines 342-343: triphenyphospane - ? triphenylphospine?;

line 349: ...phosphanium bromide - ...phosphonium bromide;

line 447: 105 CFU.mL-1 - use correct symbol instead of dot;

Author Response

REVIEWER 1

Dear reviewer, thank you for taking the time to review our manuscript. We appreciate all your suggestions and effort for the text improvement. Find below a point-by-point response to your valuable comments.

The manuscript "Caffeic acid after derivatization revealed selective anticancer, strong antimicrobial, and promising anti-Acanthamoeba activity" by M. Lukáč is devoted to the caffeic acid derivatization and bioactivity testing. The authors describe a few novel phosphonium salts derived from caffeic acid in 3 steps. The main part of the presented material is a results of different assays (amoebocidal, antibacterial, antifungal, cytotoxic effects and surface activity).

  1. The authors use underived caffeic acid as a control in all assays. I guess it's not correct: to evaluate the effect and correctness of experimental data we should see the real positive controls, specific for each type of testing. Cytotoxic agent (like doxorubicin), antibacterial, antifungal etc. data should be added to make the conclusions about the compounds' activity. As a result, the discussion section should be extensively rewritten.

For cytotoxicity tests, we used cisplatin as a reference compound and added it in table 2.  We also added parts that might need to be added to the results and discussion. Please see the highlighted text. Regarding antimicrobial and anti-acanthamoeba testing, we used well-characterized collection strains of microorganisms, preserved except in the Czech Collection of Microorganisms and also in the other renowned microbial culture collections (amongst them in the ATCC) – see the list of the strains with their recommended use in the clinical diagnostical microbiology. Staphylococcus aureus CCM 4223 / ATCC 29213 MSSA; antimicrobial susceptibility quality control strain; Staphylococcus aureus CCM 4750 / ATCC 43300 MRSA; methicillin susceptibility reference strain; Enterococcus faecalis CCM 4224 / ATCC 29212 antimicrobial susceptibility quality control strain; Pseudomonas aeruginosa CCM 3955 / ATCC 27853 antimicrobial susceptibility quality control strain; Escherichia coli CCM 3954 / ATCC 25922 antimicrobial susceptibility quality control strain; Klebsiella pneumoniae CCM 4415 / ATCC 10031 antimicrobial susceptibility quality control strain; Proteus mirabilis CCM 7188 / ATCC 29906 assays of antimicrobial preservative quality control; Candida albicans  CCM 8267 /ATCC 90028 - reference strain for CLSI-developed antifungal susceptibility testing. The used broth micromethod for antimicrobial susceptibility testing (recommended both by the EUCAST and the CLSI) is a routinely used highly standardized assay, not require a reference antibiotic. As a negative control, wells free of microorganisms are used. Wells with microorganisms in the medium free of any antimicrobials are involved as a positive control. To identify a potential technical failure of the assay, three independent runs were employed; no failure was detected (already explained in the text lines 516 – 521) As well as for antiprotozoal testing. Positive control: The growth control after 24 hours (acanthamoeba in inoculum without treatment): A. lugdunensis (AcaVNAK02): 2.5 x 105 cells/mL A. quina (AcaVNAK03): 2.1 x 105cells/mL in comparing to initial inoculum 1x 105 cells/mL – this information was added into the text (lines 538 – 541).

  1. It's a very specific type of chemical modification - introduction of phosphonium group. Usually it has an aim to redirect the compounds into the mitochondria of eucariotic cells or affect the procariotic cells in similar way (membrane integrity and potential). Here I had not find any explanations why the authors did the presented transformations and what was the aim.

Caffeic acid is a well-known molecule with detailed descriptions of different pharmacological effects. The improvement by phosphanium group led to the change of crucial physical properties - into a surfactant-like molecule. This was a pilot screening for changed/improved biological effects. The cytotoxicity increased toward both eucaryotic and procaryotic cells. Anyway, as we concluded, prepared molecules are suitable within the biological activities known so far more for topical applications. We rewrote the aims to make them clearer. Please see the highlighted text.

Based on these points I recommend to reject the paper in the current form.

Comments on the Quality of English Language

The section 4. Materials and Methods should be carefully checked and improved.

We checked and improved the text again.

For example,

line 296: MgSO4 - use subscript for 4; DONE (it happened by formatting the text)

line 299: 1H NMR - use superscript for 1; DONE

line 301: 13C NMR (75 MHz, CDCl3); DONE

line 306: 7.41g (0.053 mol) potassium carbonate (7.41 g, 0.052 mol); We corrected this mistake.

lines 342-343: triphenyphospane - ? triphenylphospine?;

We corrected according: Favre H.A. and Powell W.H. Nomenclature of organic chemistry IUPAC Recomendations and Preffered Names 2013

line 349: ...phosphanium bromide - ...phosphonium bromide;

We unified it in the whole text according to: Favre H.A. and Powell W.H. Nomenclature of organic chemistry IUPAC Recomendations and Preffered Names 2013

line 447: 105 CFU.mL-1 - use correct symbol instead of dot; DONE – all checked/corrected.

Reviewer 2 Report

Comments and Suggestions for Authors

Dear Editor,
I have carefully reviewed the manuscript titled "Caffeic acid after derivatization revealed selective anticancer, strong antimicrobial, and promising anti-Acanthamoeba activity" and I would like to provide my comments and suggestions for improvement.

Overall, the study presented in the manuscript is highly valuable and addresses an important area of research.

The authors have successfully synthesized four caffeic acid phosphonium derivatives and evaluated their cytotoxic, antibacterial, antifungal, and amoebicidal activities in vitro. The results demonstrate that the synthesized derivatives exhibit significantly stronger biological activities compared to caffeic acid itself. The findings have potential implications for the development of novel selective antitumor and antimicrobial agents.

Here are my specific comments and suggestions:

  1. Title: The current title is informative but can be further improved to reflect the main findings of the study. Consider revising it to something like: "Enhanced Biological Activities of Caffeic Acid Phosphonium Derivatives: Potential as Selective Antitumor and Antimicrobial Agents."

  2. Abstract: The abstract provides a concise summary of the study; however, it would benefit from some minor revisions. Firstly, it would be helpful to briefly mention the specific cell lines used for cytotoxicity testing. Additionally, it would be useful to include the concentrations of the synthesized derivatives and caffeic acid that were tested. Also, it would be valuable to specify the potential applications of the synthesized derivatives and further research.

  3. Introduction: The introduction provides a good overview of caffeic acid and its pharmacological activities. However, it would be beneficial to include a brief rationale for the synthesis of phosphonium derivatives, highlighting the potential benefits and advantages of modifying caffeic acid in this manner.

  4. Materials and Methods: The section describing the synthesis of caffeic acid phosphonium derivatives is well-detailed and appears to be reproducible. However, it would be helpful to include the chemical structures of the synthesized derivatives for clarity.

  5. Results: The results section is comprehensive and clearly presents the data obtained from the biological evaluations. It would be beneficial to include representative dose-response curves for the cytotoxicity assays to visualize the enhanced potency of the synthesized derivatives compared to caffeic acid.

  6. Discussion: The discussion provides a thorough analysis of the results and their implications. However, it would be valuable to discuss the potential mechanisms of action underlying the enhanced biological activities of the phosphonium derivatives. Additionally, it would be helpful to address the limitations and challenges associated with the practical application of these derivatives, such as stability and bioavailability.

  7. Conclusion: The conclusion is well-written and appropriately summarizes the main findings of the study. Consider adding a sentence that highlights the significance of the study and its potential impact on the development of new therapeutic agents.

Author Response

Dear reviewer, thank you for taking the time to review our manuscript. We appreciate all your suggestions and effort for the text improvement. We implemented all your recommendations into our manuscript. Please see our replies below.

Title: The current title is informative but can be further improved to reflect the main findings of the study. Consider revising it to something like: "Enhanced Biological Activities of Caffeic Acid Phosphonium Derivatives: Potential as Selective Antitumor and Antimicrobial Agents."

Thank you for your suggestion. We changed the title.

Abstract: The abstract provides a concise summary of the study; however, it would benefit from some minor revisions. Firstly, it would be helpful to briefly mention the specific cell lines used for cytotoxicity testing. Additionally, it would be useful to include the concentrations of the synthesized derivatives and caffeic acid that were tested. Also, it would be valuable to specify the potential applications of the synthesized derivatives and further research.

The abstract has been improved according to your recommendations. Please see the abstract.

Introduction: The introduction provides a good overview of caffeic acid and its pharmacological activities. However, it would be beneficial to include a brief rationale for the synthesis of phosphonium derivatives, highlighting the potential benefits and advantages of modifying caffeic acid in this manner.

The introduction was improved – please see the improved aims of the study.

Materials and Methods: The section describing the synthesis of caffeic acid phosphonium derivatives is well-detailed and appears to be reproducible. However, it would be helpful to include the chemical structures of the synthesized derivatives for clarity.

The structure is presented in Fig. 1. For better resolution, we added the structure in supporting information (supplementary material) (Figure S1)

Results: The results section is comprehensive and clearly presents the data obtained from the biological evaluations. It would be beneficial to include representative dose-response curves for the cytotoxicity assays to visualize the enhanced potency of the synthesized derivatives compared to caffeic acid.

We added caffeic acid in Fig 3 to see the difference between CA and CAPs.

Discussion: The discussion provides a thorough analysis of the results and their implications. However, it would be valuable to discuss the potential mechanisms of action underlying the enhanced biological activities of the phosphonium derivatives. Additionally, it would be helpful to address the limitations and challenges associated with the practical application of these derivatives, such as stability and bioavailability.

As you recommended, we extended the discussion on potential mechanisms of action and the limitations in usage. Please see the highlighted text.

Conclusion: The conclusion is well-written and appropriately summarizes the main findings of the study. Consider adding a sentence that highlights the significance of the study and its potential impact on the development of new therapeutic agents.

Thank you, we rewrote the conclusion. Please see the highlighted text.

Reviewer 3 Report

Comments and Suggestions for Authors

The paper titled "Caffeic acid after derivatization revealed selective anticancer, strong 2 antimicrobial, and promising anti-Acanthamoeba activity," contributed by S. B. Fialová et al., reports the preparation of caffeic acid phosphonium salts (CAPs) and the test of the prepared compounds for their cytotoxic, antibacterial, antifungal, and amoebicidal activities in vitro. I have several critical concerns about the publication of this manuscript.

Major Points:

Supplementary Materials:

The manuscript does not adhere to the journal's recommendation of including Supplementary Materials. It is crucial to provide essential data such as images and spectral information for the prepared compounds to facilitate readers in assessing the quality and purity of the synthesized materials.

Comparison with Acetylated Caffeic Acid:

The manuscript draws a comparison with acetylated caffeic acid (compound I) and references other papers where NMR data were obtained in DMSO-d6 instead of CDCl3 as recently these data were published (https://doi.org/10.1016/j.carres.2022.108683).

Methodology for Ester Synthesis:

The manuscript utilizes an unconventional methodology for synthesizing esters of acetylated caffeic acid, deviating from the standard esterification approach. The authors should justify this choice and address concerns related to poor nucleophilicity. A more thorough explanation or comparison with established methodologies is necessary.

Novelty of Compounds:

The 4 esters of acetylated CA and 4 phosphonium salts prepared are unknow compounds, as it is possible to check. Given the unconventional strategy to prepare , it is imperative to provide a complete characterization of the compounds before any biological evaluation. This should include mass spectrometry data (both nominal and exact mass), purity information, and images of spectral data and/or HPLC data.

Conclusion

Due to the concerns regarding the preparation of CAPs, I strongly recommend non-acceptance of the manuscript in its current form.

Author Response

Dear reviewer, thank you for taking the time to review our manuscript. We appreciate all your suggestions and effort for the text improvement.

The paper titled "Caffeic acid after derivatization revealed selective anticancer, strong 2 antimicrobial, and promising anti-Acanthamoeba activity," contributed by S. B. Fialová et al., reports the preparation of caffeic acid phosphonium salts (CAPs) and the test of the prepared compounds for their cytotoxic, antibacterial, antifungal, and amoebicidal activities in vitro. I have several critical concerns about the publication of this manuscript.

Major Points:

Supplementary Materials:

The manuscript does not adhere to the journal's recommendation of including Supplementary Materials. It is crucial to provide essential data such as images and spectral information for the prepared compounds to facilitate readers in assessing the quality and purity of the synthesized materials.

 We added Supporting information (supplementary material) with all information.

Comparison with Acetylated Caffeic Acid:

The manuscript draws a comparison with acetylated caffeic acid (compound I) and references other papers where NMR data were obtained in DMSO-d6 instead of CDCl3 as recently these data were published (https://doi.org/10.1016/j.carres.2022.108683).

 We add a comparison of NMR spectra to the manuscript (reference No. 47)

Methodology for Ester Synthesis:

The manuscript utilizes an unconventional methodology for synthesizing esters of acetylated caffeic acid, deviating from the standard esterification approach. The authors should justify this choice and address concerns related to poor nucleophilicity. A more thorough explanation or comparison with established methodologies is necessary.

 This type of preparation of esters is not very common. The methods are discussed in: Otera J. Esterification. Methods, Reactions, and Applications Wiley-CH 2003 ISBN 3-527-30490-8. The reference (22) regarding the preparation of aliphatic lactones is included in the manuscript.

Novelty of Compounds:

The 4 esters of acetylated CA and 4 phosphonium salts prepared are unknow compounds, as it is possible to check. Given the unconventional strategy to prepare, it is imperative to provide a complete characterization of the compounds before any biological evaluation. This should include mass spectrometry data (both nominal and exact mass), purity information, and images of spectral data and/or HPLC data.

The NMR spectra of all phosphanium salts are attached in the supplementary information. Three spectral methods characterized all compounds. The purity was confirmed by the presence of one peak in 31P NMR spectra.

Round 2

Reviewer 1 Report

Comments and Suggestions for Authors

- The protocols described in the section "4.2.2 Antibacterial and antifungal activity testing" are non-standard. For example:

The test was performed in 100 ul volumes in microtiter plate wells with a final bacterial concentration of 105 CFU·mL-1 - According current version of EUCAST recommendations it should be 5x105 CFU·mL-1.

As a result we can not evaluate the results in abscence of any positive controls (antibiotics). These data should be added.

I also recommend to make table 3 more readable - using the abbreviations with long bottom text looks strange. Please, add the full strain names in the main body. The table could be transposed.

- References 49&50 should be for EUCAST recomendations (links) or for equivalent ISO standard (docs).

The manuscript becomes more readable but still neeeds serious revision.

Comments on the Quality of English Language

line 22: spectra - spectrum;

line 68: that displayed with - rewrite for clarity;

line 324: phosphonium - phosphanium?

lines 396-397: triphenyphospane - triphenylphosphine?

line 403: 1H - 1H

Author Response

Dear reviewer,

Thank you very much for taking the time to review the revision. We´re thankful for the additional comments.

The protocols described in the section "4.2.2 Antibacterial and antifungal activity testing" are non-standard. For example:

The test was performed in 100 ul volumes in microtiter plate wells with a final bacterial concentration of 105 CFU·mL-1 - According current version of EUCAST recommendations it should be 5x105 CFU·mL-1.

thank you for the reminder on the final inoculum size in the microdilution MIC tests - we corrected a formal mistake.

As a result we can not evaluate the results in abscence of any positive controls (antibiotics). These data should be added.

As we used bacteria collection, it is a standard. Different bacteria are sensitive to different antibiotics. To avoid testing all possible antibiotics, we used caffeic acid as a standard, as the antimicrobial activity of caffeic acid is well documented (https://doi.org/10.1021/acs.jafc.0c07579). We compare the antimicrobial activity of CAP derivatives with solo caffeic acid, focusing on the difference in the final effect.

Usually, there is no requirement for positive control in the case of a microdilution test. A positive control is necessary for the diffusion test:

Mohamed EAA, Muddathir AM, Osman MA. Antimicrobial activity, phytochemical screening of crude extracts, and essential oils constituents of two Pulicaria spp. growing in Sudan. Sci Rep. 2020 Oct 13;10(1):17148. doi: 10.1038/s41598-020-74262-y. PMID: 33051571; PMCID: PMC7555867.

Other papers on caffeic acid derivatives also mentioned caffeic acid as a standard:

Kępa M, Miklasińska-Majdanik M, Wojtyczka RD, Idzik D, Korzeniowski K, Smoleń-Dzirba J, Wąsik TJ. Antimicrobial Potential of Caffeic Acid against Staphylococcus aureus Clinical Strains. Biomed Res Int. 2018 Jul 15;2018:7413504. doi: 10.1155/2018/7413504. PMID: 30105241; PMCID: PMC6076962.

Ma CM, Abe T, Komiyama T, Wang W, Hattori M, Daneshtalab M. Synthesis, anti-fungal and 1,3-β-D-glucan synthase inhibitory activities of caffeic and quinic acid derivatives. Bioorg Med Chem. 2010 Oct 1;18(19):7009-14. doi: 10.1016/j.bmc.2010.08.022. Epub 2010 Aug 13. PMID: 20813534.

Ginovyan M, Petrosyan M, Trchounian A. Antimicrobial activity of some plant materials used in Armenian traditional medicine. BMC Complement Altern Med. 2017 Jan 17;17(1):50. doi: 10.1186/s12906-017-1573-y. PMID: 28095835; PMCID: PMC5240328.

I also recommend to make table 3 more readable - using the abbreviations with long bottom text looks strange. Please, add the full strain names in the main body. The table could be transposed.

DONE. We added the full strain names.

- References 49&50 should be for EUCAST recomendations (links) or for equivalent ISO standard (docs).

Both links were added. Thank you for the notice; the library system didn´t insert the whole reference.

The manuscript becomes more readable but still neeeds serious revision.

Comments on the Quality of English Language

line 22: spectra - spectrum; corrected

line 68: that displayed with - rewrite for clarity; corrected

line 324: phosphonium - phosphanium? corrected

lines 396-397: triphenyphospane - triphenylphosphine? corrected

line 403: 1H - 1H corrected 

Reviewer 3 Report

Comments and Suggestions for Authors

The revised version of the manuscript, now titled "Caffeic Acid Phosphanium Derivatives: Potential Selective Antitumor and Antimicrobial Agents," has addressed some of the previously raised concerns, yet certain issues persist:

Supplementary Material Inconsistency:

Supplementary material has been appropriately incorporated for Caffeic Acid Phosphanium derivatives (CAPs); however, it remains absent for the Caffeic Acid esters (CA esters).

31P NMR Purity Criterion:

The assertion that 31P NMR can serve as a purity criterion is not fully convincing. The 31P NMR spectra of the CAPs exhibit a single signal, but only the CAP12 spectrum encompasses a sufficiently broad window to preclude the presence of other phosphorus species, such as the starting material PPh3.

It is noteworthy that the chemical shift window commonly described in literature for triphenylalkylphosphonium 31P spectra (between 20 and 26 ppm, doi.org/10.1016/j.tet.2023.133321) is comparable to the value for OPPh3 (29.19 ppm, doi.org/10.1002/adsc.202300351).

Given the aforementioned considerations, I maintain reservations regarding the purity of the tested compounds.

Author Response

Dear reviewer,

Thank you very much for taking the time to review the revision. We´re thankful for the additional comments.

The revised version of the manuscript, now titled "Caffeic Acid Phosphanium Derivatives: Potential Selective Antitumor and Antimicrobial Agents," has addressed some of the previously raised concerns, yet certain issues persist:

Supplementary Material Inconsistency:

Supplementary material has been appropriately incorporated for Caffeic Acid Phosphanium derivatives (CAPs); however, it remains absent for the Caffeic Acid esters (CA esters).

We included the NMR data in supporting information.

31P NMR Purity Criterion:

The assertion that 31P NMR can serve as a purity criterion is not fully convincing. The 31P NMR spectra of the CAPs exhibit a single signal, but only the CAP12 spectrum encompasses a sufficiently broad window to preclude the presence of other phosphorus species, such as the starting material PPh3.

It is noteworthy that the chemical shift window commonly described in literature for triphenylalkylphosphonium 31P spectra (between 20 and 26 ppm, doi.org/10.1016/j.tet.2023.133321) is comparable to the value for OPPh3 (29.19 ppm, doi.org/10.1002/adsc.202300351).

Given the aforementioned considerations, I maintain reservations regarding the purity of the tested compounds.

Regarding the purity, we added the HPLC-DAD of all four compounds. Please see the supporting information. The method is included in the chapter Materials and Methods. These additional analyses were provided over a longer time (explanation for some minority peaks).

Round 3

Reviewer 1 Report

Comments and Suggestions for Authors

The minor corrections could be added during proofreading. Now the manuscript could be accepted for publication.

Comments on the Quality of English Language

- References 49&50 should be for EUCAST recomendations (links) or for equivalent ISO standard (docs).

Both links were added. Thank you for the notice; the library system didn´t insert the whole reference.

Now it's refs. 50&51 - please, improve it manually if you can not to do this using reference manager.

Author Response

Dear Reviewer,

Thank you very much. The reference manager rewrote it again by refreshing it. The links were added manually.

Reviewer 3 Report

Comments and Suggestions for Authors

Contrarily to the statment presented on line 435 the mass spectra are not preented on the  supporting information. 

Author Response

Dear Reviewer,

Thank you very much. The mistaken statement was corrected.